# Altered Brain Reactivity to Food Cues in Undergraduate Students with Disordered Eating Behaviors

**DOI:** 10.3390/biomedicines12081656

**Published:** 2024-07-25

**Authors:** Joao C. Hiluy, Isabel A. David, Isabela Lobo, Filipe Braga, Thayane Fernandes, Naiane Beatriz Ferreira, Maria Francisca F. P. Mauro, Jose C. Appolinario

**Affiliations:** 1Obesity and Eating Disorders Group, Institute of Psychiatry, Federal University of Rio de Janeiro, Rio de Janeiro 22290-140, Brazil; mffpmauro@gmail.com (M.F.F.P.M.); jotappo@gmail.com (J.C.A.); 2Laboratory of Behavioral Neurophysiology, Physiology and Pharmacology Department, Biomedical Institute, Federal Fluminense University, Niteroi 24210-130, Brazil; isabeldavid@id.uff.br; 3Psychobiology Group, Integrated Morphology Laboratory, Institute of Biodiversity and Sustainability, Federal University of Rio de Janeiro, Macae 27965-045, Brazil; isabela.neuro@gmail.com (I.L.); thayaneferreiraf@gmail.com (T.F.); naianebeatrizf@gmail.com (N.B.F.); 4Psychobiology Group, Institute of Medical Sciences, Federal University of Rio de Janeiro, Macae 27930-560, Brazil; filipebraga82@yahoo.com.br

**Keywords:** eating disorders, hyper-palatable foods, electroencephalography, neurophysiology

## Abstract

Purpose: A growing body of evidence has shown that electroencephalography (EEG) is an interesting method of assessing the underlying brain physiology associated with disordered eating. Using EEG, we sought to evaluate brain reactivity to hyper-palatable food cues in undergraduate students with disordered eating behavior (DEB). Methods: After assessing the eating behaviors of twenty-six undergraduate students using the Eating Attitudes Test (EAT-26), electroencephalographic signals were recorded while the participants were presented with pictures of hyper-palatable food. The current study used a temporospatial principal component analysis (PCA) approach to identify event-related potential (ERP) responses that differed between DEB and non-DEB individuals. Results: A temporospatial PCA applied to the ERPs identified a positivity with a maximum amplitude at 347 ms at the occipital–temporal electrodes in response to pictures of hyper-palatable food. This positivity was correlated with the EAT-26 scores. Participants with DEB showed reduced positivities in this component compared with those without DEB. Conclusion: Our findings may reflect greater motivated attention toward hyper-palatable food cues in undergraduate students with DEB. These results are an important step toward obtaining a more refined understanding of specific abnormalities related to reactivity to food cues in this population.

## 1. Introduction

There is a growing body of evidence indicating that individuals with eating disorders display altered cognitive responses that are mainly elicited by eating disorder-related cues, such as pictures of food and bodies [1,2,3]. Furthermore, individuals with disordered eating behaviors (DEBs) who do not fulfill the full criteria for an eating disorder diagnosis may exhibit altered patterns of cognitive processing associated with these cues, especially food images. These behaviors may include restrictive eating, binge eating, self-induced vomiting, laxative misuse, or irregular or inflexible eating patterns [4]. As a matter of fact, the current literature, although heterogeneous, indicates that individuals with DEBs may present several changes in cognitive processing, particularly in relation to food stimuli [5].

The evaluation of food-related cognition can be useful for understanding the underlying brain mechanisms related to DEB. One method used to evaluate brain reactivity to visual food cues is the event-related potential (ERP) technique, which consists of measuring brain waveforms derived from electroencephalographic signals. This provides information about how brain processes unfold over time in response to visual stimuli [6]. ERP components can reflect a variety of neural functions, such as visual processing, and higher-order attention processes, including emotional and motivation-based attention [7].

Previous research suggests that more palatable foods elicit larger ERP amplitudes than less palatable foods [5]. Furthermore, studies that evaluated higher-order attention processes in individuals with disturbed eating behaviors by assessing ERPs such as the Late Positive Potential (LPP), the Early Posterior Negativity (EPN), and P300 suggest that hyper-palatable foods capture greater attention than non-hyper-palatable foods or neutral stimuli in this population. Thus, these findings show that individuals can be sensitive to food stimuli in their environment and therefore allocate more attention to food cues.

These disturbed eating behaviors could be more prevalent in some populations. For instance, student status predicted an increased risk of subsequent binge eating disorder and bulimia nervosa in females [8]. Additional data presented in a systematic review that evaluated thirty-three studies that analyzed the risk of eating disorders in undergraduate students indicated that students in courses in the health area, especially nutrition, present a higher risk of eating disorders [9]. However, despite these studies pointing to this increased risk, we found only one study assessing neurophysiology in this population. A study published in 2012 showed that in this population, impulsivity and inhibitory control deficits are positively associated with responses to external food cues. In response to negative emotional states, these deficits result in food-related decision-making based on taste preferences without consideration of health values [10]. It is important to better evaluate this group not only from a clinical point of view but also in terms of the neurobiological correlates involved. With this perspective in mind, the present study aimed to investigate whether ERPs associated with the emotional processing of hyper-palatable food cues were different in individuals with and without DEBs in a sample of undergraduate students. Our hypothesis was that individuals with DEBs would exhibit a stronger attentional bias toward high-calorie food cues than individuals without DEBs, which would be indicated by varying ERP component magnitudes, including P300, the Early Posterior Negativity (EPN), and the Late Positive Potential (LPP), across individuals with and without DEBs. 

This is an expanding area in which there is a need for exploratory and confirmatory studies. Thus, observable, measurable, and manageable biomarkers and behavioral events in the context of neurobiological and genetic research are increasingly needed, and data on undergraduate students are scarce.

## 2. Materials and Methods

### 2.1. Participants

A convenience sample comprising thirty-three participants recruited from different undergraduate courses at the Federal University of Rio de Janeiro, Macaé Campus, between October 2018 and July 2019, was gathered. Students were recruited through an invitation issued by the researchers in university classrooms to participate in a laboratory study on eating behavior and electroencephalography. Due to EEG data acquisition problems, the final sample consisted of 26 participants. The criteria for inclusion in the present study were (1) being an undergraduate student at the Federal University of Rio de Janeiro, (2) being between 18 and 30 years old, (3) being a native Portuguese speaker, and (4) reporting normal or corrected vision. The exclusion criteria were (1) being vegetarian or vegan (this was necessary since there were images of foods of animal origin among the stimuli, which could have generated different responses to the components of interest in vegetarians [11]), (2) continuously using medication or drugs that act on the central nervous system (due to possible interference with the electroencephalographic recording), and (3) reporting the occurrence of any psychiatric or neurological disease diagnosed by a physician. The experimental procedures were approved by the Research Ethics Committee of the Federal University of Rio de Janeiro, Macaé Campus. After recruitment, sociodemographic data (sex, age, and drug abuse) were collected, as were weight (Kg) and height (meters) data.

### 2.2. Disordered Eating Behavior

Eating behavior was assessed using the “Eating Attitudes Test-26” (EAT-26). This instrument fits into a set of self-administered questionnaires. It was developed by Garner and Garfinkel in 1979 [12], and a version was translated and validated for Brazil in 2004 [13]. It is a screening tool that does not have diagnostic power, but it has proved to be able to screen new cases and to estimate the severity of eating symptoms. Many studies have used the EAT-26 to identify populations at high risk of developing AN and BN [14,15,16]. The questions are formulated using Likert-type scales with 6 answers. The scores range from 3 to 0 as follows: Always (3), Often (2), Sometimes (1), Rarely (0), Almost Never (0), and Never (0). Although the cutoff point established for the risk of eating disorders is traditionally 21 [17], several studies have indicated that a lower cutoff point can be better for screening EDs among samples of high-risk individuals, as demonstrated by Orbitello et al. They evaluated 845 subjects and found that a cutoff value of EAT-26 = 11 improved the sensitivity of the EAT-26 in a high-risk setting and led to a reduction in the false-negative rate [18]. Therefore, we used an EAT-26 cutoff value of 11 points.

### 2.3. Stimulus and Procedure

The stimulus was composed of pictures depicting 32 hyper-palatable ultra-processed foods with high caloric content and low nutritional value, such as industrialized foods rich in fat, trans fat, sugar, or sodium [19,20,21]. There were two exemplary pictures of each food (e.g., cake picture 1 and cake picture 2). Thus, there were 64 food pictures in total. The 64 pictures were divided into 2 sets of 32 pictures, with each set containing the same foods but different exemplary pictures of these foods (e.g., cake 1 was allocated to set 1, and cake 2 was allocated to set 2). The 2 sets of food pictures did not differ in hedonic valence (t(62) = −034, *p* = 0.73) or emotional arousal (t(62) = 0.14, *p* = 0.89) based on previously obtained normative ratings [19]. The presentation of the two sets of pictures was balanced across the DEB and non-DEB participants. This manipulation reduced the possibility that interindividual variability in processing specific sensory aspects of the pictures (such as brightness and contrast) would account for the results [22].

This experiment took place in a prepared room with dim ambient light and sound attenuation at the Psychobiology Laboratory located on the premises of the Macaé Science and Technology Institute. During the experiments, the participants were positioned in front of a computer screen with their heads supported on a forehead/chin supporter so that the distance between the computer monitor and their eyes was maintained at 57 cm. A microcomputer running E-prime^®^ Software (Psychology Software Tools Inc., Pittsburgh, PA, USA) timed the presentation of the stimuli on the computer screen and delivered triggers (event markers) related to the onset of the food pictures for the event-related potential analysis.

The experiment was composed of 32 trials. Figure 1 shows the sequence of events in each trial. A black screen appeared for 100 ms to begin the test. Then, a fixation point (center cross) was displayed with a jittered time window (1000–1100 ms) to encourage the participants to keep their eyes on the screen. First, the participants read text containing information about the food depicted in the picture that would be presented, such as its nutritional content and methods of storage. The text was used to engage the participants’ attention on the picture and appeared for 6000 ms. Then, a picture of a hyper-palatable food with high caloric value and low nutritional value was presented at the center of the screen for 4000 ms. The participants were instructed to observe the picture attentively (passive viewing task). After the picture was removed, the participants were instructed to perform an affective rating task, which consisted of rating the food depicted in the picture on two dimensions of emotion (hedonic valence and emotional arousal) by completing a computer version of the Self-Assessment Manikin (SAM) scale [23]. The scores ranged from one to nine. The manikins displayed expressions ranging from “smiling-happy” (score = 9) to “frowning-unhappy” (score = 1) for the valence dimension, and their expressions spanned the emotional arousal spectrum, with a score of 1 for “relaxed-sleepy” and a score of 9 for “excited wide-eyed” (score = 9). The participants also rated their intention to consume the food depicted in the picture, which varied from 1 (no intention) to 9 (maximum intention). Each rating lasted approximately 5000 ms, and the participants used the numbers on the right side of the keyboard to perform the ratings (scores of 1 to 9). The software E-prime 2.0 recorded their keyboard responses (Psychology Software Tools Inc., Pittsburgh, PA, USA). EEG signals were collected throughout the experiment, and the segments of interest in the EEG signals occurred when the food pictures were presented (during the passive viewing task).

### 2.4. Electrophysiological Assessment

#### 2.4.1. EEG Recording and Analysis

The EEG data were recorded at a sampling rate of 600 Hz through an EMSA amplifier (Lynx Tecnologia Eletrônica Ltda, São Paulo, Brazil) coupled to 23 electrodes positioned on the scalp according to the 10-20 international system (FPz, Fp1, Fp2, Fz, F3, F4, F8, F7, Cz, C3, C4, T7, T8, P7, P8, Pz, P3, P4, Oz, O1, O2, M1, and M2). The EEG analysis was performed using EEGLAB [24], a free toolbox for Matlab (MathWorks, Natick, MA). The Cz channel was used as a reference during signal recording, and the signal was re-referenced to the average reference (the average of the signals at all EEG electrodes) during pre-processing (offline). The impedance was kept below 5 kΩ. The data were filtered offline using 0.1 Hz high-pass and 30 Hz low-pass second-order Kaiser digital filters. The rejection of blink-related artifacts was performed through independent component analysis [25] using the Infomax ICA algorithm in EEGLAB [24]. It took a visual examination of the ICA scalp maps to confirm the eye blink components’ proximity to the ocular area and established waveform characteristics before a maximum of two were removed from the data. The frontal lateral electrodes (F7 and F8) were examined to track saccadic eye movements, and the segments that contained them were not included in the analysis. The EEG data were epoched from 200 ms before food picture onset (baseline period) to 1000 ms after food picture onset and then averaged. Epochs containing deviations larger than 100 μV relative to the baseline for any of the electrodes were rejected. Through a visual inspection of the data, the validity of this automated procedure was further examined. Per participant, the epoch rejection rate was not more than 20%. The artifact-free epochs that were time-locked to the onset of food pictures were baseline-adjusted, averaged across the trials, and then averaged across the DEB and non-DEB participants to obtain the grand average. Principal component analysis (PCA) was applied to the obtained event-related potentials.

#### 2.4.2. Principal Component Analysis

Given that multiple ERP components may be elicited by the emotional content of food pictures (such as the LPP, the EPN, the Slow Positive Wave (SPW), and P300) [5] and that there is substantial spatial and temporal overlap between these ERPs, we applied a temporospatial principal component analysis (PCA) to the data. The PCA promoted data reduction through a statistical decomposition of the ERPs, providing better distinctions between the latent ERP components and a more objective, data-driven estimate of the ERPs [26]. The ERP components of interest were quantified using a temporospatial PCA, and all of the following data and analyses were based on this approach. PCA is a factor-analytic statistical approach that is widely used as an effective linear reduction method for multivariate ERP data. It can be used to disentangle and identify latent ERP components. Previous studies applied PCA to examine emotion-related ERP components [27,28,29]. The PCA analysis was performed in two steps, beginning with a temporal Promax rotation (with a covariance relationship matrix and Kaiser weighting) followed by a spatial Infomax rotation [30,31]. For the temporal PCA, all time points in each trial were used as variables, and the observations included all 26 participants and 20 electrodes (all electrodes except M1, M2, and FPz). For the spatial PCA, the 20 electrodes were used as variables, and all participants and temporal factors were used as observations. The temporal PCA yielded 11 factors, and 2 spatial factors were extracted for each temporal factor. This yielded a total of 22 temporospatial factor combinations. Of these, 15 factors accounted for more than 0.5% of the variance and were retained for further examination.

The PCA factors derived from the ERP data were represented by a “peak latency” (the time point with the greatest absolute voltage), “a peak negative electrode” (the electrode with the greatest negative voltage), and “a peak positive electrode” (the electrode with the greatest positive voltage). We selected the PCA factors for the statistical analysis according to a priori knowledge about the ERP components relevant to the experimental design [31,32]. Thus, the PCA factors of interest were related to ERP components that were previously associated with food-evoked emotion (such as the EPN, P300, and the LPP) [5]. In this vein, we selected PCA factors for analysis that were in close temporal and spatial proximity to the LPP, P300, and the EPN (electrodes with the largest negative or positive amplitudes at central–parietal–occipital sites and peak latencies between 200 and 700 ms) [33,34]. Following these criteria, of the 11 PCA factors that were initially selected, 4 remained: 347O1P, 347C4N, 622O2P, and 250O2N. The factor names were based on their relevant temporal (i.e., the peak latency) and spatial (i.e., the channel of interest containing the greatest positive or negative voltage) information [28]. For instance, “347O1P” means that this PCA component peaked 347 ms after the onset of the food picture, with a maximal positive amplitude at the O1 electrode.

### 2.5. Statistical Analysis

For the brain measures, the variable of interest compared between the DEB and non-DEB groups was the maximal positive or negative voltages (measured in μV) of the PCA factors (347O1P, 347C4N, 622O2P, and 250O2N) obtained for each participant. For the behavioral measures, ratings of different food pictures were combined to obtain a rating score for each participant. Thus, the variable of interest was the pictures’ mean ratings for each participant for each behavioral measure (valence, arousal, and intention-to-consume ratings). The variables of interest were then split for the DEB and non-DEB groups for intergroup comparisons. For both the behavioral and brain variables, independent-sample t-tests were applied for intergroup (DEB vs. non-DEB) comparisons. When the assumption of normality was not met (Shapiro–Wilk test), a non-parametric test (Mann–Whitney U test) was performed to compare the DEB and non-DEB groups. The analyses were performed independently for each PCA factor and behavioral measure (valence, arousal, and intention-to-consume ratings). The effect sizes of significant differences were estimated using Cohen’s *d* values (Student’s *t*-test) or rank biserial correlations (Mann–Whitney test). The significance level adopted in all analyses was α = 0.05.

## 3. Results

The total sample was composed of 26 individuals (10 biological science students, 9 pharmacy students, 4 nursing students, 2 engineering students, 2 nutrition students, and 2 students with missing data). The DEB group was composed of eleven participants (73.3% females) with a mean age of 21.14 years (SD = 2.79). The non-DEB group was 72.7% females and had a mean age of 22.73 years (SD = 5.75). The two groups did not display significant differences regarding sex, body mass index (BMI), or age (see Table 1). Regarding the food pictures, the valence (t(24) = −0.21, *p* = 0.83), arousal (U = 73.50, *p* = 0.66), and intention-to-consume (t(24) = 0.62, *p* = 0.54) ratings did not differ between the DEB and non-DEB participants.

After performing the temporospatial PCA, we selected four factors for analysis: two occipital positivities peaking at 347 ms and 622 ms (347O1P and 622O2P) and two negativities (occipital and central) peaking at 250 ms and 347 ms (250O2N and 347C4N). The occipital positivity peaking at 347 ms (347O1P) showed higher values for the non-DEB participants than for the DEB participants. The other comparisons between the DEB and non-DEB participants did not reach statistical significance (Table 2). The factor 347O1P was more positive at the occipital–temporal electrodes (T5/P7, T6/P8, O1, O2, and Oz) and reached its maximum value at O1 (See Figure 2A). The grand average waveforms of the ERPs obtained before the PCA for the occipital–temporal channels are presented in Figure 2B.

## 4. Discussion

Our study adds to the scarce literature on the neurobiology of disordered behavior in undergraduate students. In our study, using a temporal–spatial PCA applied to event-related potential waveforms, we found that participants with DEBs presented significantly reduced posterior positivity compared to individuals without DEBs when exposed to pictures of hyper-palatable foods. The disparity in the responses to these pictures occurred at occipital–temporal electrodes with a maximum amplitude 347 ms after the picture onset, as captured by the factor 347O1P.

A previous study that evaluated electrocortical correlates in undergraduate students was carried out by Blechert et al. They classified participants as having high or low levels of emotional eating. Thus, ERPs elicited in response to high-calorie food stimuli were evaluated. The objective was to assess how habitual emotional eating styles interact with negative (relative to neutral) emotional states during food cue processing. A significant group difference was then observed at occipito-parietal sites, with more positive LPP amplitudes in individuals with high levels of emotional eating. These amplitudes emerged at around 200 ms and extended across the whole LPP time range [35]. Emotional eating can be understood as one form of DEB. Thus, our study went beyond emotional eating to include other disturbed eating behaviors.

Some characteristics of the factor 347O1P, such as the occipital–temporal localization and the peak at around 350 ms, are consistent with previous reports on the EPN. The EPN is a waveform with a positive or negative polarity that is more negative (or less positive) for emotion-laden stimuli compared to neutral stimuli [36,37,38]. Previous studies have shown that engaging attention with motivationally salient (emotion-laden) stimuli is associated with an EPN between 200 and 400 ms [39,40,41]. Thus, the reduced posterior positivity found for the undergraduate students with DEBs in comparison to those without DEBs may reflect increased motivated attention to food cues in the DEB participants.

Our findings are in line with previous neurophysiological studies performed in individuals with eating disorders. A study by Blechert et al. observed a reduced positivity at posterior electrodes in the time range of 220 to 310 ms (similar to the one found here) in response to pictures of low-calorie and high-calorie foods relative to neutral pictures in individuals with eating disorders, whereas this was only seen in healthy control participants for the high-calorie pictures [3]. Additionally, another study that assessed N200 (an ERP that reflects conflict detection during the regulation of successful behavior) to compare responses to food and non-food images among individuals with binge eating found a significantly enhanced negativity in response to food stimuli (chocolate pictures) [42]. By collating data related to brain responses to food stimuli among individuals with eating and weight disorders, a systematic review performed in 2019 reported that individuals with binge eating disorder showed an enhanced attentional response to food cues associated with P300, the LPP, and N200 compared to healthy controls [1].

Contrary to the literature, no significant differences between the DEB and non-DEB groups were found regarding late temporal windows (such as 622O2P). For instance, Svaldi et al. found evidence that in women with binge eating disorder, high-calorie food pictures elicited larger LPP and SPW amplitudes (the Late Positive Potential and the Slow Positive Wave, respectively), with both ERPs occurring after >500 ms [43]. Another study that diverged from our findings found no differences in the EPN (an ERP that reflects the differential processing of emotions compared to neutral stimuli). The authors presented highly salient stimuli (food pictures) to women with bulimia nervosa, and there was no difference when compared to healthy controls [44]. These differences between our findings and the previous results may be due to sample heterogeneity or even the different stimuli/paradigms that were adopted.

It is well established that emotional pictures elicit an enhanced parietal positivity beginning around 300 ms after stimulus presentation and that intrinsic (stimulus-related) and extrinsic (context-related) factors modulate the magnitudes of these responses [45]. Therefore, our study suggests that the difference in the cognitive processing of hyper-palatable food pictures in individuals with disordered eating behavior occurs from the beginning of the stimulus presentation, and it is possible that it primarily indicates an initial allocation of attention to a motivationally salient (food) stimulus. Thus, before higher cognitive processing, the stimulus is already understood differently neurophysiologically. This may represent a neurobiological finding correlated with disordered eating behavior. Individuals with eating disorders may be included in these findings since it has already been shown that these individuals may exhibit different cognitive processing of food stimuli, such as early sensory processing, attention allocation, valence processing, cognitive/attentional processing (initial cognitive responses), and impaired allocation of motivational attribution or emotional attention [2].

This study had some limitations that should be pointed out. There was no neutral condition (regarding food characteristics). The EPN is usually considered a deflection at posterior electrodes in response to emotional stimuli relative to neutral stimuli. Thus, the interpretation of the factor 347O1P as being associated with the EPN should be considered with caution. However, although the absence of a neutral condition may affect the interpretation of the results, it does not invalidate the differences found for this factor between the individuals. Future studies may confirm the association between the EPN and the factor 347O1P. Although the weight and height data were self-reported, there is extensive research reporting that there are no significant differences between self-reported and measured weight and height data [46]. Therefore, in our assessment, this did not represent a significant limitation. Another point that should be noted is the lack of an a priori sample size calculation. However, a vast portion of the ERP literature has established robust and replicable interaction effects with similar numbers of participants [47]. Additionally, the obtained Cohen’s d effect size was considered large (Cohen’s d > 0.8), and there is a positive relationship between reliability and effect sizes in between-group ERP studies [48]. Finally, all participants were undergraduate students. Consequently, care should be taken when generalizing the findings to other populations. Nevertheless, this study has important strengths, including the use of standardized and validated pictures and the application of a validated instrument.

EEG may therefore provide a window to assess the neurophysiological behavior of individuals with disturbed eating behaviors [2]. Studies using this technique may be useful for diagnostic and treatment purposes and for understanding the interaction between the more automatic and controlled processing of emotional stimuli. EEG data could be useful to assess whether changes correspond to a diagnostic marker or are already present in individuals with disturbed eating behaviors. Additionally, for treatment proposals, EEG studies could perform tasks intended to modify attentional processes [49], making it possible to modulate patterns of dysfunctional behaviors related to food stimuli. Another way to use EEG is through neurofeedback, which could train individuals to actively and voluntarily regulate their neural activity in response to real-time feedback via an EEG interface [50].

In conclusion, undergraduate students with DEBs presented significantly reduced posterior positivity compared to those without DEBs when exposed to pictures of hyper-palatable ultra-processed foods. We suggest that future studies be conducted with standardized food cues, using neutral stimuli as a control, and with an experimental design to evaluate EEG components throughout the entire duration of cognitive processing, from sensory input to a motor outcome, using both the earliest ERPs and later ones. Thus, EEG-based studies may, in the future, help identify individuals at higher risk of developing eating disorders in a defined sample, such as undergraduate students, and even be used to monitor eating symptoms when present in a high-risk population.

## Figures and Tables

**Figure 1 biomedicines-12-01656-f001:**
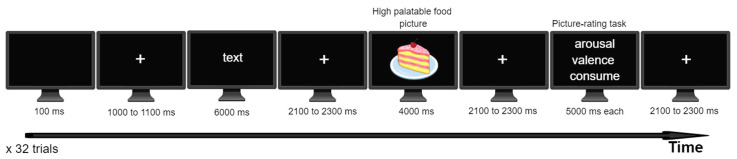
The sequence of events during a trial. The participants viewed pictures depicting hyper-palatable foods preceded by text that engaged their attention on the pictures. After viewing a food picture (such as a piece of cake, for illustrative purposes), the participants performed a rating task in which they provided ratings of valence, arousal, and intention to consume for the food pictures.

**Figure 2 biomedicines-12-01656-f002:**
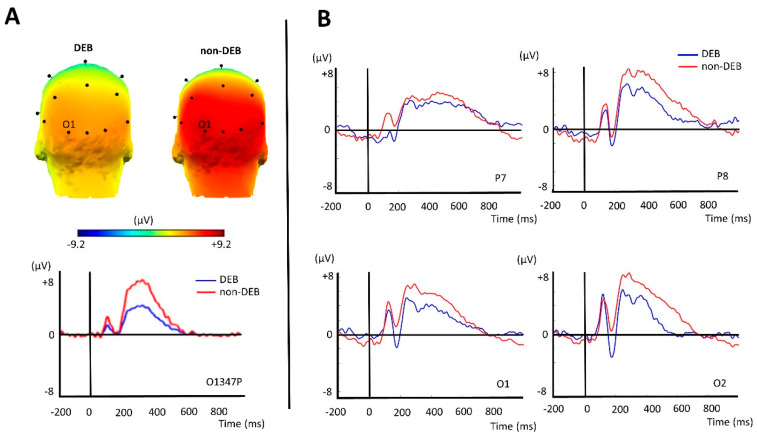
(**A**) The top panel shows a topographic map of the scalp (3D head, posterior view). The voltage distributions of the factor O1347P across the posterior electrodes are shown for the DEB and non-DEB participants. The colormap changes from blue to red as the voltages (µV) become more positive over the scalp. The maximal positive voltage occurred at the O1 electrode. The DEB participants presented reduced positivity compared to the non-DEB participants. The bottom panel shows the waveform for the factor O1347P, with its maximal peak amplitude around 347 ms. (**B**) Event-related potential waveforms from four occipital–temporal electrodes (P7/T5, P8/T6, O1, and O2). The pattern of the ERP waveforms goes in the same direction as that of the factor O1347P (presented in Figure 2A), showing a reduced positivity in the DEB participants compared to the non-DEB participants.

**Table 1 biomedicines-12-01656-t001:** Characteristics of the sample of undergraduate students.

	EAT-26 Group	
	DEB	Non-DEB
Sample size (n)	11	15
Age, mean (SD)	22.73 (±2.80)	21.14 (±5.75)
Sex (% female)	72.7%	73.3%
Mean BMI (kg/m^2^)	21.83 (±2.43)	21.88 (±4.85)

BMI: body mass index; DEB: disordered eating behavior; SD: standard deviation.

**Table 2 biomedicines-12-01656-t002:** Comparison between the DEB and non-DEB participants for the selected PCA factors.

PCA Factors	Peak Amplitude (µV) (Mean (SD))		Test	t or U	*p*-Value	Effect Size
	DEB (*n* = 11)	Non-DEB (*n* = 15)				
347O1P	3.94 (3.31)	7.39 (4.58)	Student	2.12	0.04	0.84
347C4N	−0.98 (1.91)	−0.38 (2.50)	Student	0.66	0.52	-
622O2P	−0.371 (2.68)	2.23 (3.51)	Mann–Whitney	112	0.13	-
250O2N	−0.10 (3.98)	−0.13 (2.58)	Student	−0.022	0.98	-

DEB: disordered eating behavior; PCA: principal component analysis; SD: standard deviation.

## Data Availability

The data presented in this study are available on request from the corresponding author due to ethical reasons.

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
