# Peer review of "Altered Brain Reactivity to Food Cues in Undergraduate Students with Disordered Eating Behaviors"

_biomedicines, 2024, doi:10.3390/biomedicines12081656_

Round 1

Reviewer 1 Report

Comments and Suggestions for Authors

This study investigated whether event-related brain potentials (ERPs) in response to viewing pictures of hyper-palatable, high-calorie foods differed between undergraduate students with and without disordered eating behaviors (DEB). EEG data was recorded while participants viewed food images, and principal component analysis (PCA) was applied to the ERP data. The results showed that students with DEB displayed a significantly reduced posterior positivity around 347ms after picture onset compared to those without DEB..The findings assumed to suggest neurophysiological evidence of altered food cue reactivity in undergraduates exhibiting disordered eating patterns. This is an original study that could contribute existing literature. I have several comments on article;

1.      The author provided some reasons for focusing on undergraduate students like they are at risk. However they should present more robust data to justify their interest in this sample and declare what gaps the current study fills.

2.      In addition the introduction should present a clear hypothesis statement or the main research question.

3.     The details on participant recruitment are lacking (lines 80-82). How were the sample recruited? What were recruitment methods? Please clarify.

4.     The sample size of 26 participants seems quite small, especially when split into two groups. Besides using a convenience sample from a single university limits the generalizability of the findings. These are important limitations of study that should be mentioned and discussed. Authos should add power analysis results to justify this sample size.

5.     Inclusion/exclusion criteria should be clearer and needs justification (lines 83-87). For example, why exclude vegetarians/vegans or those on medications? These criteria seem arbitrary. Please explain.

6.     The procedure for the experiment should have more details. For example state how long each trial lasted, what the participants were instructed to do during the trial, or how the data was collected and stored. Add details on EEG data processing procedures like artifact removal methods beyond ICA, filtering parameters, etc. (lines 151-166).

7.     The use self-reported measures for eating without clinical assessments is an important limitation that should be added and discussed.

8.     The statistical analysis section should include more details. For example; detailing the variables used with tests,

9.     Discussion is overall satisfactory except for limitations. Limitations of the study design, methods, or sample should also be addressed. For example, the small sample, lack of control condition, order effects, and generalizability should be mentioned and discussed along with others stated above.

Reviewer 2 Report

Comments and Suggestions for Authors

The article seeks to evaluate brain reactivity through EEG to hyperpalatable food cues in university students with eating disorders (ED). It is a paper with an adequate design, a sufficient sample, a careful and elaborate methodology, with pertinent analysis, and original results.

Author Response

Thank you for the comment